# Towards the Optimization of Apodized Resonators

**DOI:** 10.3390/mi16050511

**Published:** 2025-04-27

**Authors:** Ana Valenzuela-Pérez, Carlos Collado, Jordi Mateu

**Affiliations:** Signal Theory and Communications (TSC) Department, Universitat Politècnica de Catalunya (UPC), 08034 Barcelona, Spain; ana.valenzuela@estudiantat.upc.edu (A.V.-P.); jordi.mateu-mateu@upc.edu (J.M.)

**Keywords:** BAW resonator, film bulk acoustic resonator, solidly mounted resonator, spurious modes, apodization

## Abstract

Bulk Acoustic Wave (BAW) resonators are essential components in modern RF communication systems due to their high selectivity and quality factor. However, spurious resonances caused by Lamb wave mode propagation along the in-plane directions degrade the filter performance. Traditional Finite Element Method (FEM) simulations provide accurate modeling but are computationally expensive, especially for arbitrarily shaped resonators and solidly mounted resonators (SMRs), whose stack of materials is composed of many thin layers of different materials. To address this, we extend a previously published model (named the Quasi-3D model), which employs the Transmission Line Matrix (TLM) method, enabling efficient simulations of complex geometries with more precise meshing. The new approach allows us to simulate different geometries, and we will show several apodized geometries with the aim of minimizing the lateral modes. In addition, the proposed approach significantly reduces the computational cost while maintaining high accuracy, as validated by FEM comparisons and experimental measurements.

## 1. Introduction

BAW resonators are key components in modern communication systems because of their high selectivity and high quality factor (Q) values. However, they face significant challenges related to lateral modes, that is, Lamb waves that propagate along the lateral directions of the resonator [1]. These modes can introduce spurious resonances within the pass-band, degrading the performance of BAW-based filters [2,3,4].

To model the effects of lateral modes, FEM techniques have traditionally been used. Although these simulations offer precise results, they present a high computational cost, especially when they are required to model solidly mounted resonators (SMRs) with arbitrary in-plane geometries [5]. With the objective of reducing this computational load, alternative models have been proposed, such as analytical solutions for two-dimensional simulations [6,7] and approaches based on the TLM method [8].

The Quasi-3D model developed in [9] represents a significant advancement that enables the analysis of resonators with a varied range of in-plane geometries, without the necessity of performing complete 3D FEM simulations. However, the model shown in [9] was limited to relatively simple electrode shapes and meshing techniques that may not completely capture the structural complexity of some resonators. In this work, we extend the previous model, enabling the simulation of more complex geometries by means of more precise meshes, which better adapt to the in-plane geometry of the resonator.

To mitigate the unwanted effects of lateral modes, there are mainly two types of methods. The first consists of building a frame around the edge of the top electrodes, i.e., a border ring (BR) [10]. Then, by adjusting its width and thickness, suppression of the spurious resonances is achieved. The second method is called apodization. It involves designing the top electrode in an asymmetric way [11], so that there are no parallel edges in the geometry. By doing so, the standing lateral waves are smeared out between the electrode edges and the lateral modes are minimized, avoiding their appearance in the electrical response. It is worth mentioning that most of the high-performance resonators make use of both techniques, border ring and apodization. This work describes the TLM model as a very efficient simulation method for designing apodized resonators optimized to minimize the lateral modes.

This article is structured as follows. The next section briefly presents the Quasi-3D model and details the modifications made with respect to the previous work published in [9] to improve its accuracy and adaptability to complex geometries. Subsequently, we validate the model implementation by comparing with FEM methods the simulations of a pentagonal Film Bulk Acoustic Resonator (FBAR), where we apply different meshing strategies to evaluate their impact on accuracy and convergence. We then describe an optimized five-sided SMR design intended to minimize the lateral modes. Following this, we compare the efficiency of our approach against conventional FEM simulations in terms of computational time, considering different meshing techniques. Finally, we discuss the experimental validation results and conclude with key takeaways and future research directions.

## 2. Quasi-3D Model with Adaptive Meshing

The Quasi-3D model introduced in [9] leverages the TLM method to model wave propagation along the lateral dimensions of the resonator, using precomputed key dispersive properties of the Lamb wave modes. The model eliminates the need for simulating the thickness direction of the structure and discretizes only the lateral dimensions. This is achieved by constructing a nodal Y-matrix, where each node represents a discretized section of the resonator and is interconnected through transmission lines that emulate the lateral wave propagation, as shown in Figure 1.

One of the main limitations of the original Quasi-3D model is its reliance on uniform rectangular-shaped meshes, which restricts its ability to accurately represent complex resonator shapes. In this work, we extend the model by introducing a more flexible approach that is more suitable for simulating non-orthogonal geometries, such as pentagonal or apodized resonators. The following section details these modifications and their implementation.

### 2.1. Unit Cell

In [9], the in-plane geometry of the resonator is discretized using a uniform mesh of squared cells, where each cell can have different side lengths dx and dy. Additionally, the current is injected into the nodes of the discretized mesh. Since all cells have the same surface area, this uniform mesh is well suited to simulate orthogonal structures, such as rectangles or squares. However, this approach restricts the range of geometries that can be accurately represented, as the mesh is only suitable for simple orthogonal shapes. Although it is possible to simulate more complex structures, such as the trapezoidal resonator described in [9], it introduces unavoidable errors. The only way to mitigate these errors is by increasing the density of the mesh, which in turn raises computational costs.

To address this limitation, we propose an adaptive mesh that better conforms to the contour of the geometry, as illustrated in Figure 2a. Unlike the uniform mesh, the new approach employs cells that are neither uniform nor strictly orthogonal, requiring adjustments in the simulation methodology. The first key modification is the placement of the current injection points. In the uniform mesh, injecting current at the vertices leads to poorly defined surface areas per node. To resolve this, we propose injecting the current at the centroid of each cell instead. As a result, the model employs two overlapping meshes: the original one (black), which discretizes the resonator’s geometry, and a secondary mesh (blue) that represents the transmission lines and the current injection nodes (red).

### 2.2. Dispersion Curve

Before detailing the implementation of the adaptive meshing approach, it is useful to briefly revisit the fundamental dispersion properties of Lamb waves.

Lamb waves are formed at the interfaces of a piezoelectric sheet as a result of the coupling between the longitudinal and shear partial waves. The phase constant β of a Lamb wave propagating in the lateral dimension (*x*-axis) through a plate of finite thickness (*z*-axis) must fulfill the Rayleigh–Lamb equation [12]: (1)$$\frac{\omega^{4}}{V_{T}^{4}}=4\cdot q^{2}\cdot \beta^{2}\cdot \left(1-\frac{p\cdot tan\;\left(\frac{p\cdot t}{2}+\theta \right)}{q\cdot tan\;\left(\frac{q\cdot t}{2}+\theta \right)}\right),$$
where $p^{2}=\frac{\omega^{2}}{V_{L}^{2}}-\beta^{2}$ and $q^{2}=\frac{\omega^{2}}{V_{T}^{2}}-\beta^{2}$. $V_{L}$ and $V_{T}$ are the phase velocities of the longitudinal and transverse waves, respectively, and ω is the angular frequency. The asymmetric modes correspond to $\theta =\frac{\pi }{2}$, while the symmetric ones correspond to θ=0. We are only interested in the second case since, as explained in [1], only these modes couple on the resonator’s electrical response.

Although the Rayleigh–Lamb equation provides the fundamental relation governing the phase constant of Lamb waves in a finite-thickness plate, its complexity makes direct implementation impractical for efficient modeling. To simplify the characterization of lateral modes in BAW resonators, we employ an approximation based on an analogy with electromagnetic waveguides. This approach assumes that a set of standing wave patterns, defined in one in-plane direction, propagate in the perpendicular direction with a dispersive phase constant. The dispersion behavior of the Thickness Extensional ($TE_{1}$) mode, for Type I resonators, can then be approximated by the following expression [13]: (2)$$\beta =\frac{2\cdot \pi }{c_{lamb}}\cdot {\left(f^{r}-f_{001}^{r}\right)}^{\frac{1}{r}},$$
where r≈2, $c_{lamb}$ is the estimated wave velocity, *f* is the operating frequency, and $f_{001}$ the cut-off frequency of the $TE_{1}$ mode, also known as the piston mode. Here, ${\left(f^{r}-f_{001}^{r}\right)}^{\frac{1}{r}}$ represents the dispersion relation’s frequency dependence, where *f* and $f_{001}$ are first raised to the power of *r* before subtraction and root operations. As shown in [13], this simple expression can accommodate fairly well the dispersive curve obtained by 2D-FEM simulations of the cross-section of a resonator, or even the dispersive curve that can be inferred from measurements of simple test resonators, as performed in [13]. Note that in addition to the cut-off frequency, there are only two parameters that define the curve: $c_{lamb}$ and *r*.

### 2.3. Characteristic Impedance

Since the cells in our new mesh are no longer orthogonal or uniform, the method to calculate the characteristic impedance described in [9] must be adjusted. The characteristic impedance for a traveling wave in an acoustic transmission line relates the force $F_{z}$ and the particle velocity $v_{z}$ [8] as follows: (3)$$F_{z}=Z_{0}\cdot v_{z},$$(4)$$F_{z}=\int_{V}\frac{\partial T_{xz}}{\partial x}dV=T_{xz}\cdot A_{tb},$$
where $A_{tb}$ is the cross-sectional area of the resonator and $T_{xz}$ is the tangential stress. This formulation assumes that the wave propagation direction in the transmission line is perfectly aligned with the normal vector to $A_{tb}$, which was valid for the uniform mesh, but does not hold in our case. As shown in Figure 2b, the transmission line direction $\hat{x}$ forms an angle with the normal vector $\hat{n}$ to $A_{tb}$. This misalignment affects the computation of force transmission and requires a correction. To properly account for it, the force $F_{z}$ must be projected onto the actual propagation direction of the wave, leading to the following expression: (5)$$F_{z}=T_{xz}\cdot A_{tb}\cdot \left(\hat{x}\cdot \hat{n}\right),$$
where $\hat{n}$ is defined outward from the cell. Consequently, incorporating this projection and following the development in Section 2 of [9], the new expression for the characteristic impedance is given by(6)$$Z_{0}=\frac{\beta \cdot c_{55}\cdot A_{tb}}{\omega }\cdot \left(\hat{x}\cdot \hat{n}\right).$$

Here, $c_{55}$ is one of the constants of the stiffness tensor of the piezoelectric, and β is the phase constant in (Equation 2). In the model, the force applied by the electric potential is represented as a distributed current source, and the resulting admittance and impedance of the transmission lines in Figure 1 are calculated by matrix inversion, as in [9].(7)$$Z_{0}^{\prime }=\frac{a_{kZ}}{Z_{0}},$$

To improve the adjustment of the electro-acoustic coupling, the impedance of each transmission line in the Y-matrix is scaled by a factor $a_{kz}$. This factor is empirically fine-tuned, since the characteristic acoustic impedance of (10) is that of an isolated thin sheet of piezoelectric and must be modified to account for a more complex multilayer structure as a real resonator is. For the transmission lines in the other in-plane direction, the unitary vector should be replaced accordingly. The impedance of the dispersive transmission line is given by(8)$$Z_{sxi}=Z_{0i}^{\prime }\cdot sinh\left(\gamma \cdot dx_{i}\right)$$(9)$$Z_{pxi}=Z_{0i}^{\prime }\cdot \frac{1}{tanh\left(\gamma \cdot \frac{dx_{i}}{2}\right)}$$(10)$$Z_{pij}=\frac{Zpxi\cdot Z_{pyj}}{Zpxi+Z_{pyj}},$$
where i,j={1,2} are as shown in Figure 1. $dx_{i}$ is the distance between the two connecting nodes, and γ=α+jβ is the complex propagation constant, with α being the attenuation. To ensure that the particle velocity is zero in the boundaries of the domain, we short-circuit the terminations of the mesh.

### 2.4. Electric Impedance

Having established the modifications required for accurately computing the characteristic impedance in our adapted mesh, we now describe the numerical formulation that allows solving for the input impedance of the resonator. This formulation is based on the nodal admittance matrix approach.

When an electric potential (*U*) is applied across the resonator electrodes [14,15], it induces a uniform force ($F_{z}$) due to the piezoelectric effect. As discussed in [9], this force can be equivalently represented as a distributed current source ($I_{source}$) at each node of the discretized structure. Therefore, the current injected in the node *i* is as follows: (11)$$I_{source,i}=\frac{e\cdot dx_{i}\cdot dy_{i}}{t}\cdot U,$$
where *e* is the piezoelectric constant, $dx_{i}\cdot dy_{i}$ accounts for the surface of the cell *i*, and *t* is the thickness between the electrodes. The electrical behavior of the resonator is described by a nodal admittance matrix of size $N_{t}\times N_{t}$, where $N_{t}$ denotes the total number of nodes in the structure. Solving the linear system in (Equation 12) allows us to obtain the nodal voltages (particle velocity), $V_{j}$, which in turn determine the resulting current flowing out of the electrodes.(12)$$\sum_{j=1}^{N_{t}}Y_{ij}\cdot V_{j}=I_{source,i}$$

From here, one obtains $V_{j}$, which as described in [9] represents the velocities, $v_{z}$, at each node. Finally, the electric current flowing out the resonator is calculated as(13)$$I=\frac{e}{t}\cdot \sum_{i=1}^{N_{t}}dx_{i}\cdot dy_{i}\cdot v_{z,i},$$
and the acoustic contribution to the total resonator’s admittance is found as follows: (14)$$Y_{lat}=\frac{I}{U}$$

Finally, the total impedance of the resonator is calculated using the expression presented in (Equation 15), where $C_{0}$ is the static capacitance and $R_{s}$ is the series resistance to Ohmic losses in the model.(15)$$Z_{in}=R_{s}+\frac{1}{Y_{lat}+j\omega C_{0}}$$

## 3. Simulation of a Pentagonal Resonator

With the Quasi-3D model, now extended to handle structures with non-orthogonal edges, we have simulated a pentagonal FBAR (see Figure 3). In this section, we validate the accuracy of our model by comparing its results with FEM simulations using COMSOL 6.0. This example will allow us to evaluate the computational efficiency of our approach, highlighting its advantages in terms of reduced computational cost and scalability.

We perform a simulation of a ZnO membrane with a thickness of 1.72 μm and an area of 1990 $\mu m^{2}$. To achieve more pronounced spurious modes, the losses are kept artificially low. To mesh the pentagon, since we employ quadrangular cells, we must divide the structure into five different regions and then connect them. We perform this in such a way that the overall mesh is symmetric.

To assess the sensitivity of our model to mesh refinement, we apply different meshing strategies (see Figure 4). The leftmost mesh is a uniform grid that covers the entire resonator. The other two meshes increase the mesh density at the structure’s edges and boundaries between regions while keeping the overall number of nodes constant. Ensuring well-meshed boundaries is crucial to minimize errors arising from changes in the transmission line directions. This is illustrated in Figure 5. In this figure, two adjacent cells from different regions meet. As seen in the figure, a misalignment occurs between the two regions, as the actual transmission line should follow the green path, but the model assigns it to the yellow path. If we include two new cells in the gray region, the error becomes negligible since, as the nodes become sufficiently close, both paths converge.

We employ two different meshing strategies: adaptive meshing, using Chebyshev nodes [16], and geometric meshing, based on a geometric progression. The geometric meshing approach is implemented using two techniques.

*Edge-based refinement.* The minimum spacing between nodes is enforced at the edges of the segments to ensure accurate meshing at the boundaries between regions, even for a small number of nodes. The node distribution, $x_{i}$, is(16)$$x_{i}=\left\{\begin{array}{cc} x_{i-1}+d_{min}\cdot b^{i-2} & \mathrm{for}\;i=2,\ldots ,\frac{N_{x}+1}{2}\\ w-x_{N_{x}+2-i} & \mathrm{for}\;i=\frac{N_{x}+1}{2}+1,\ldots ,N_{x}+1 \end{array}\right.,$$
where $d_{min}$ is the minimum spacing at the edges of the segment, *w* is the width of the segment, and *b* is the solutions of the polynomial (Equation 17):(17)$$p\left(b\right)=d_{min}\cdot \sum_{k=1}^{\frac{N_{x}+1}{2}}b^{k-1}-\frac{w}{2}.$$*Center-based refinement.* The maximum spacing between nodes is controlled at the center of the segments, preventing excessively large cells that could degrade accuracy. Similarly to the previous case, the node distribution is given by the following expression:(18)$$x_{i}=\left\{\begin{array}{cc} x_{i-1}+\frac{d_{max}}{2}\cdot b^{\frac{N_{x}+1}{2}-i} & \mathrm{for}\;i=2,\ldots ,\frac{N_{x}+1}{2}\\ w-x_{N_{x}+2-i} & \mathrm{for}\;i=\frac{N_{x}+1}{2}+1,\ldots ,N_{x}+1 \end{array}\right.,$$
where $d_{max}$ is the maximum spacing between nodes. The *b* coefficients are now given by (Equation 19):(19)$$p\left(b\right)=\frac{d_{max}}{2}\cdot \sum_{k=1}^{\frac{N_{x}+1}{2}-1}b^{k-1}-\frac{w}{2}.$$

Using the FEM simulation, we adjust the dispersion curve parameters to $c_{lamb}=6700$, r=1.995, and $f_{001}=1.7875$ GHz, the scaling factor of the impedance $a_{kz}=1.35$, the attenuation α=2000, and the piezoelectric constant e=1.32. The adjusted simulation is shown in Figure 6.

In Figure 7, we plot the Mean Square Error (MSE) of each mesh as a function of the total number of nodes in the mesh, $N_{t}$. The MSE is computed between consecutive simulations, progressively increasing the number of nodes. The criterion we have set to consider that convergence has been achieved is MSE≤15.

The graph shows that, for a given $N_{t}$ ($N_{t}>125$), the adaptive mesh achieves the lowest MSE, converging (MSE<15) at $N_{t}=6125$. This indicates that the adaptive approach effectively refines the mesh in regions where higher accuracy is required, leading to improved computational efficiency. The uniform mesh exhibits the slowest convergence, and with respect to geometric meshing strategies, better convergence is achieved when the maximum node separation is set at the center of the interval. As $N_{t}$ increases, the geometric distribution causes rapid growth in node density at the boundaries.

Based on these results, we run simulations with the best and worst meshes for comparison, but ensuring convergence. Figure 6 shows the impedance magnitude and phase: FEM simulations (blue) −MSE=12.12−, our model with a uniform mesh (red) −Nt=10125, MSE=10.94−, and with an adaptive mesh (green) −Nt=6125, MSE=9.78−.

Figure 6 shows good agreement between the FEM response and our model. Both meshing strategies yield nearly identical results, suggesting that the uniform mesh provides sufficient accuracy in most cases, but at the expense of a higher density of nodes. At higher frequencies, noticeable discrepancies emerge between our model and the FEM results. These deviations likely stem from the simple mathematical function we define for the dispersive curve of the phase constant of these modes, which cannot reproduce perfectly the FEM dispersive curve implicit in the simulations at higher frequencies.

In Figure 8a, the normalized standing wave patterns $|v_{z}|$ are shown for the first four resonant modes, using the adaptive mesh. The fundamental mode can be clearly observed. Moreover, the high symmetry of the pentagonal geometry can be seen in the other three resonant modes.

Figure 8b shows the resonant modes obtained with COMSOL. As can be seen, there is a strong agreement with the patterns obtained with the Quasi-3D models in symmetry and amplitude.

In conclusion, the Quasi-3D model was successfully extended to handle non-orthogonal structures and validated against FEM simulations, showing good agreement. Different meshing strategies were analyzed, with adaptive meshing using Chebyshev nodes proving the most efficient, achieving convergence with fewer nodes ($N_{t}=6125$) compared to uniform meshing ($N_{t}=10125$). Although both uniform and adaptive meshes yielded similar results, the adaptive approach reduced computational cost and improved mesh refinement, reducing errors. However, discrepancies emerged at higher frequencies, likely due to the simplified mathematical model used for the dispersive curve. Finally, the standing wave patterns of the first four resonant modes confirmed the expected fundamental mode and the symmetry of the pentagonal structure. These findings highlight the Quasi-3D model as a computationally efficient and accurate alternative to the FEM.

## 4. Apodization of an SMR Resonator

In this section, we aim to minimize the lateral modes in a five-sided SMR resonator (see Figure 9). In order to model it, first the material’s dispersion curve must be adjusted. This will be performed using two measured SMR resonators. Then, a pentagonal SMR will be simulated. Finally, we will proceed with the optimization.

Since lateral modes can significantly impact the performance of BAW resonators, it is essential to explore strategies to minimize them. One of the most used techniques is apodization. The main idea behind it is to design the top electrode with non-parallel edges. By modifying the geometry of the electrodes to eliminate parallel edges, the resonant paths become much longer than a symmetric-shaped resonator, making the standing waves more attenuated [17]. This extended path results in greater attenuation of spurious modes. Our goal is to illustrate how this method can be used to design an SMR five-sided resonator that minimizes the coupling of the lateral modes.

Before applying the apodization technique, it is crucial to calibrate our model to find the parameters of the dispersive curve and the other parameters previously described. To achieve this, we utilized experimental data from two reference SMR resonators. The composition of the layers is as follows: the top electrode is composed of an Al layer, another of *W* with a passivation layer of SiN at the top, the bottom electrode is composed of *W* and Al, and the Bragg reflector alternates layers of $SiO_{2}$ with *W* layers. The entire stack remains on a Si substrate. One of the resonators is square while the other is rectangular, and both have the same area of 6400 $\mu m^{2}$. Their series resonance frequency is around 2.48 GHz. Figure 10 shows the adjustment of our model, in red, to the measurements, in blue. To achieve these results, the model parameters were set to the following: $c_{lamb}=5120$, r=2.21, $a_{kz}=4.7$, α=13,500, and e=1.36.

Then, we proceeded to minimize the lateral modes. As a first step in the optimization process, we simulated a pentagonal SMR resonator, maintaining the same layer composition and surface area as in the previous case. The goal is to assess how an additional edge, completely avoiding parallel edges, diminishes the lateral modes. We used the adaptive mesh, and the results obtained are shown in Figure 11.

Notice that the lateral modes have been reduced, in comparison with the squared resonators, as expected, but there are still many well-defined modes because of the symmetry of the geometry.

To improve the frequency response of the resonator, we modify the geometry further, keeping the area constant, in order to lose the symmetries present in the pentagonal structure. An optimal design is depicted in Figure 12.

In Figure 11, we also plot (in blue) the magnitude and phase of the input impedance of the apodized resonator. To facilitate the observation of the lateral mode mitigation, we plot in Figure 13 a zoom-in of the phase of the lateral impedance for four cases: the pentagonal SMR, the apodized SMR, the square SMR, and the rectangular SMR.

As seen in the input impedance in Figure 11, the lateral modes have been minimized. This can be more clearly observed in Figure 13, where the phases of the input impedance of the pentagonal, the square, and the apodized resonators are compared. It is important to note that other layouts could lead to similar or even better results. Beyond the goodness of the tested geometry, these results highlight that this method can be used to optimize the apodization of a resonator without recurring complex 3D-FEM simulations of SMR resonators.

## 5. Computational Analysis

To evaluate the computational efficiency of our approach, we compare the execution times and degrees of freedom (DoFs) required by the FEM and the Quasi-3D model using the different meshing strategies for the pentagonal FBAR. The SMR computational cost is also introduced for three different structures, using only the adaptive mesh, since it is the one with the fastest convergence. The dispersion curve parameters (i.e., $c_{lamb}$ and *r*) are precomputed from the FEM or experimental data and reused across simulations. The execution time for 2D-FEM simulations for the FBAR stack is 10 s. This one-time cost is negligible compared to the runtime savings per type of geometry. Table 1 summarizes the results. All simulations were performed using the same number of frequencies and achieved mesh convergence. It is important to mention that with the FEM simulation, conveniently taking advantage of the symmetry in the pentagonal geometry, only half of the structure was simulated, significantly reducing the computational load. The simulations were run in a system equipped with an Intel Core i9-12900K processor (16 cores, 24 threads, 3.2 GHz base, variable boost), 64 GB of DDR4 RAM, and an NVMe SSD for storage. The system features a dual-GPU setup, including an NVIDIA T400 (4 GB) and Intel UHD Graphics.

A key observation from the table is the substantial reduction in DoFs when using the Quasi-3D model compared to the FEM. This efficiency gain arises because the FEM must resolve the resonator’s thickness direction, whereas the Quasi-3D model inherently incorporates this dimension through precomputed dispersion relations. This is why it is not possible to simulate an SMR with this geometry using the FEM, because there are many thin layers, and thus the DoF would be too high to be simulated with our available computers. Nevertheless, with our model, simple geometries such as the square or the rectangular one can be simulated in a short time. Furthermore, the pentagonal SMR converges with the same number of DoFs as the FBAR.

To further improve computational performance, we implemented parallelization strategies in our solver when computing the linear system (Equation 12). Since the system must be solved for each frequency point, we can distribute each computation of the Y-matrix across multiple threads using MATLAB R2024a. To avoid long idle times between threads, tasks are dynamically assigned upon completion. Although our resources were limited to eight cores, we analyzed the scalability of our implementation using a speed-up plot (see Figure 14).

The plot illustrates how the speed-up of the code varies with the number of threads. The blue curve represents the actual speed-up observed, while the black dashed line represents the ideal linear speed-up, where doubling the number of threads would ideally double the speed. As shown in the figure, the observed speed-up deviates from the ideal linear scaling, exhibiting sublinear behavior. This performance limitation is likely caused by synchronization overhead, which could be mitigated through more advanced parallelization techniques, such as load balancing or memory access patterns. Despite this, it is worth noting that increasing the number of threads does improve performance; therefore, the computational time could be further reduced.

## 6. Summary and Conclusions

Lateral spurious modes in BAW resonators pose a significant challenge in high-frequency applications, affecting both the resonator’s performance and its integration into RF systems. This work has presented an efficient computational model based on the TLM method to accurately predict these lateral resonances. Using precomputed dispersion curves and modeling the lateral wave propagation through discretized transmission lines, the proposed Quasi-3D approach achieves results that closely match FEM simulations while significantly reducing computational costs.

A significant improvement in our model lies in the advanced meshing techniques employed. By incorporating both uniform and adaptive meshing, with non-orthogonal cells, we achieved a higher resolution in areas of interest. The adaptive mesh refinement allowed for more accurate modeling of high-frequency responses while achieving convergence using fewer nodes. This flexibility enabled us to simulate complex electrode geometries, including apodized structures with non-parallel edges, which would be computationally almost prohibitive using traditional FEM approaches.

Another significant contribution of this work is the introduction of parallelization techniques to reduce computing time. The parallel assembly of the nodal admittance matrices and the use of optimized solvers, along with our proposed model, allowed us to reduce the simulation time by several orders of magnitude compared to FEM methods.

The accuracy of the model was validated against both FEM simulations and experimental measurements, demonstrating its ability to predict the impact of resonator geometry on lateral modes. We showed as an example an apodized design that successfully diminished the harmful effects of spurious resonances while maintaining the resonator’s fundamental frequency response, illustrating the practical effectiveness of this approach.

Beyond its direct application to FBAR or SMR resonators, the methodology introduced in this work can be extended to other electro-acoustic devices where lateral resonances must be controlled, such as SAW and XBAR resonators. Future work will focus on refining the dispersion curve in order to reduce the deviation of the prediction at high frequencies, incorporating a BR frame to apodized resonators, and developing optimization algorithms to automatically design the optimum in-plane geometry for a given stack of materials.

## Figures and Tables

**Figure 1 micromachines-16-00511-f001:**
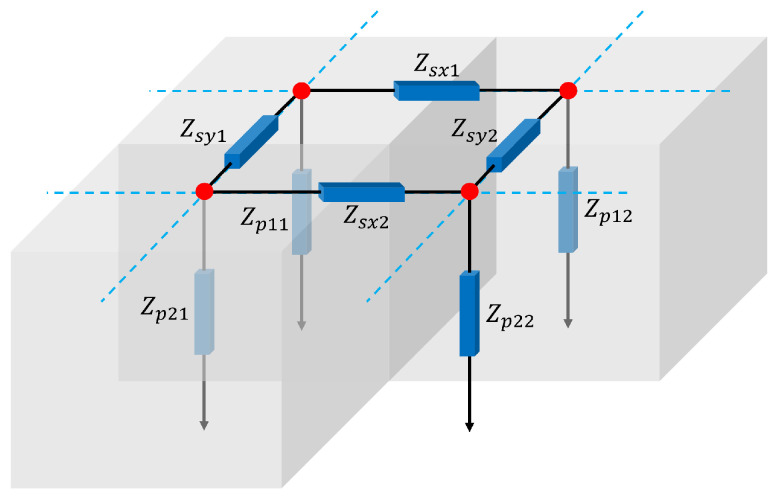
Scheme of the connection structure of the different transmission lines.

**Figure 2 micromachines-16-00511-f002:**
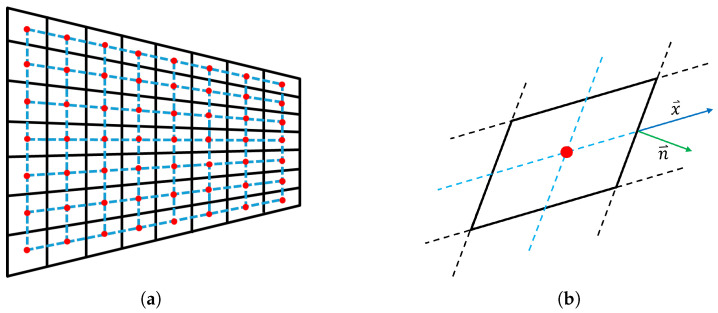
(**a**) Discretization example of a trapezoidal resonator. In black, the mesh that discretizes the geometry, in blue, the mesh containing the dispersive transmission lines, and in red, the nodes where the current is to be injected. (**b**) Scheme of a cell in the mesh, indicating the transmission line direction as well as the direction of the vector normal to the lateral surface area of the resonator.

**Figure 3 micromachines-16-00511-f003:**
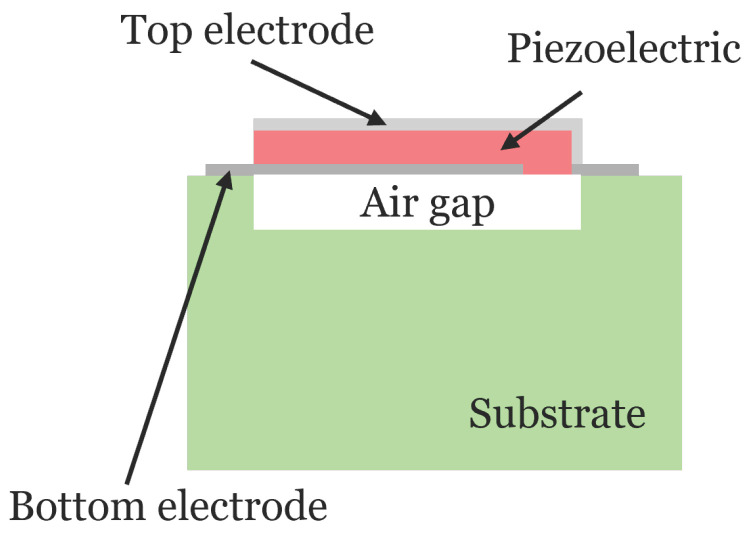
Schematic diagram of the cross-section of an FBAR resonator.

**Figure 4 micromachines-16-00511-f004:**
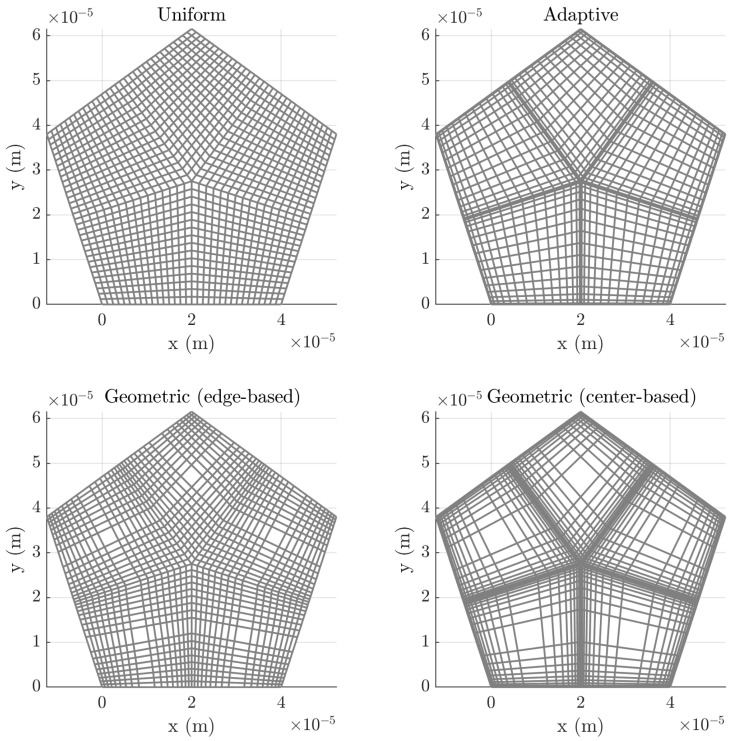
Meshed pentagonal resonator using four different meshes.

**Figure 5 micromachines-16-00511-f005:**
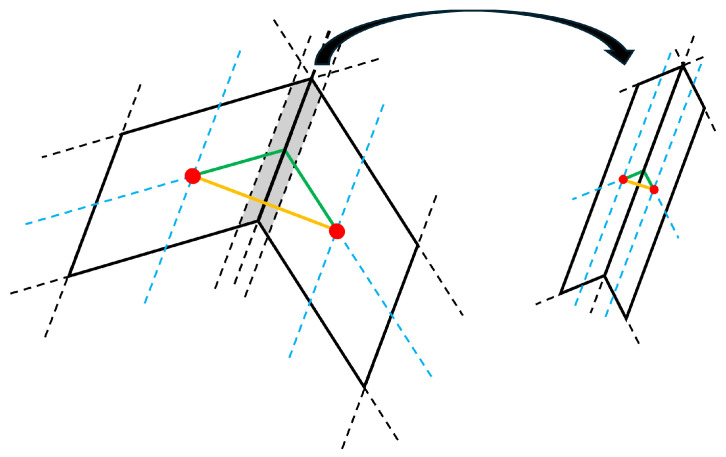
Impact of mesh refinement on transmission line alignment at region boundaries.

**Figure 6 micromachines-16-00511-f006:**
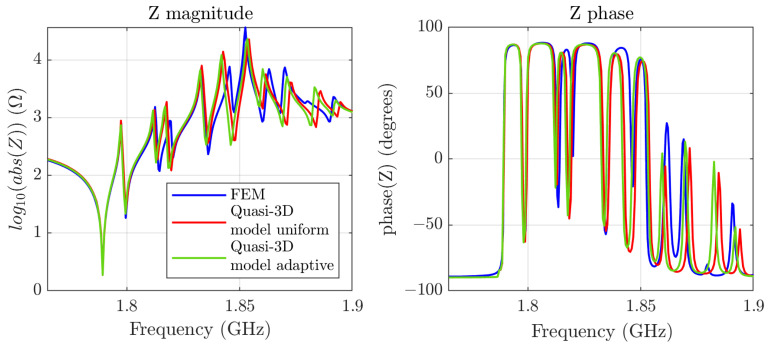
Comparison between the FEM simulations (in blue) and our model using two different meshes, a uniform one (in red) and an adaptive one (in green).

**Figure 7 micromachines-16-00511-f007:**
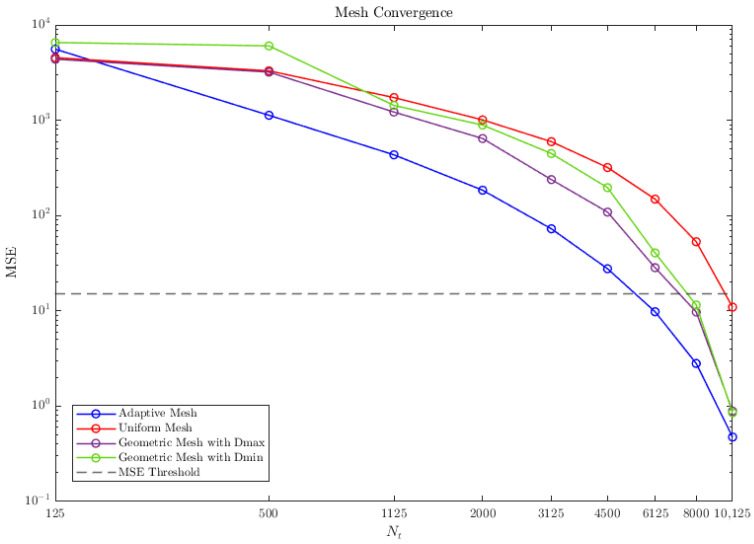
MSE as a function of the total number of nodes $N_{t}$ comparison between the adaptive mesh (in blue), the uniform mesh (in red), and the geometric meshes with the two specified settings: maximum spacing between nodes in the middle of the segment (in purple), and minimum spacing in the edges of the segment (in green).

**Figure 8 micromachines-16-00511-f008:**
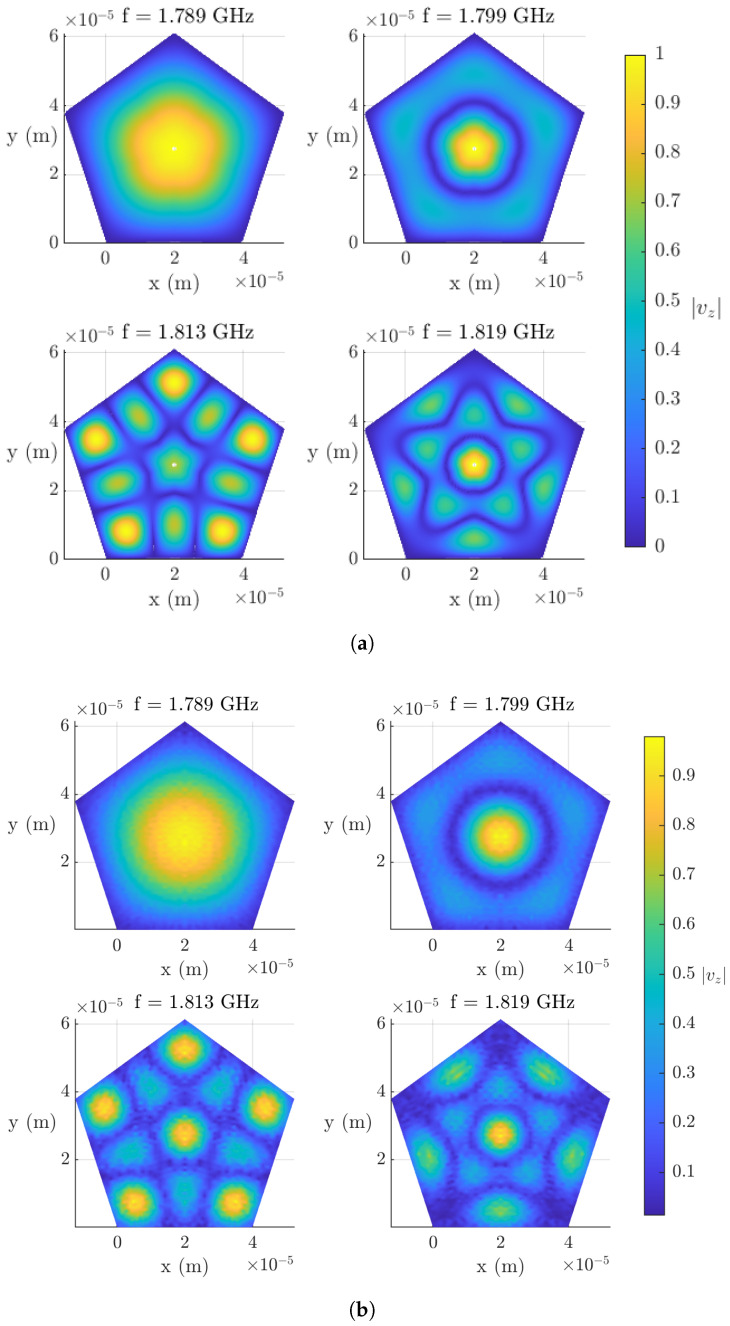
Normalized standing patterns of the resonator’s first four resonant modes: (**a**) Quasi-3D model. (**b**) COMSOL.

**Figure 9 micromachines-16-00511-f009:**
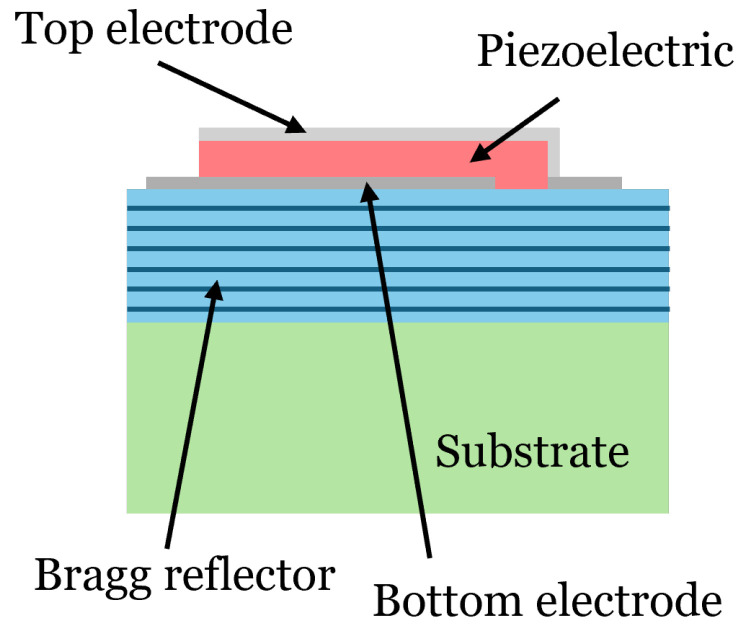
Schematic diagram of the cross-section of an SMR resonator.

**Figure 10 micromachines-16-00511-f010:**
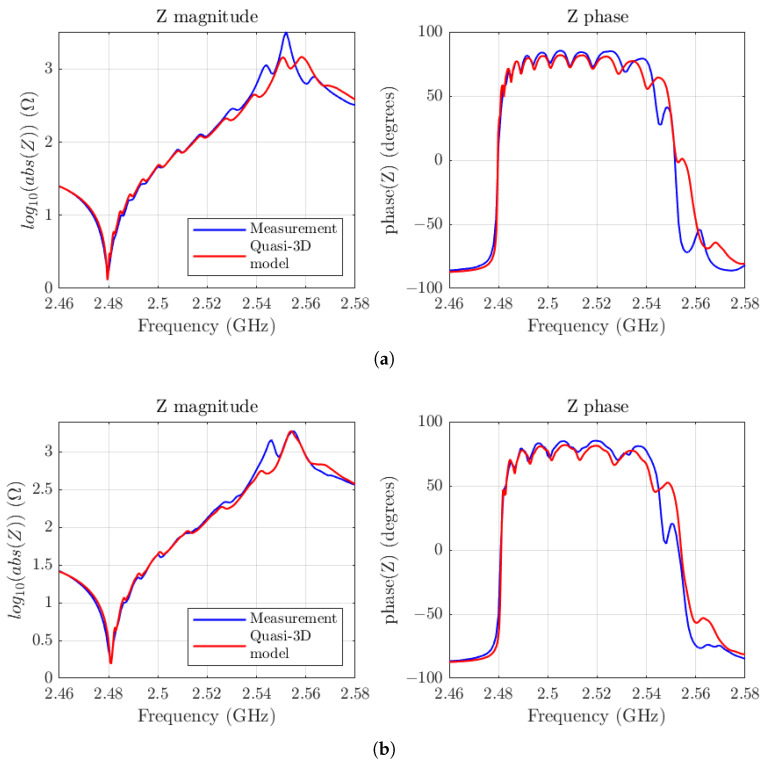
(**a**) Fitting of the Quasi-3D model, in red, with the measurement of an SMR square resonator, in blue. (**b**) Fitting of the Quasi-3D model, in red, with the measurement of an SMR rectangular resonator, in blue.

**Figure 11 micromachines-16-00511-f011:**
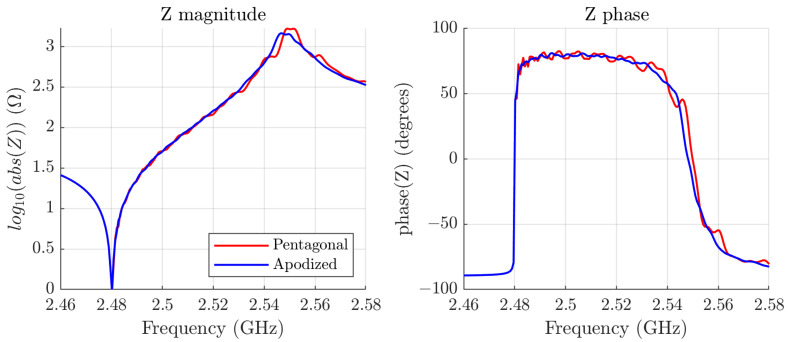
Simulation of the magnitude and phase of the pentagonal resonator (in red) and the optimized apodized resonator (in blue).

**Figure 12 micromachines-16-00511-f012:**
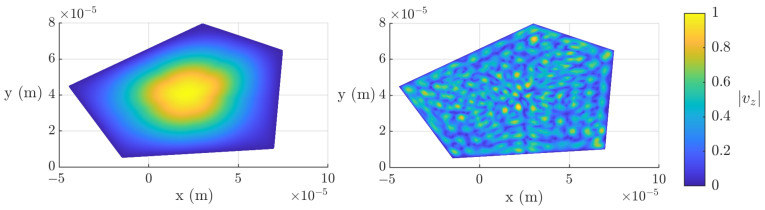
Normalized $|v_{z}|$ of two different modes for the optimized pentagonal SMR.

**Figure 13 micromachines-16-00511-f013:**
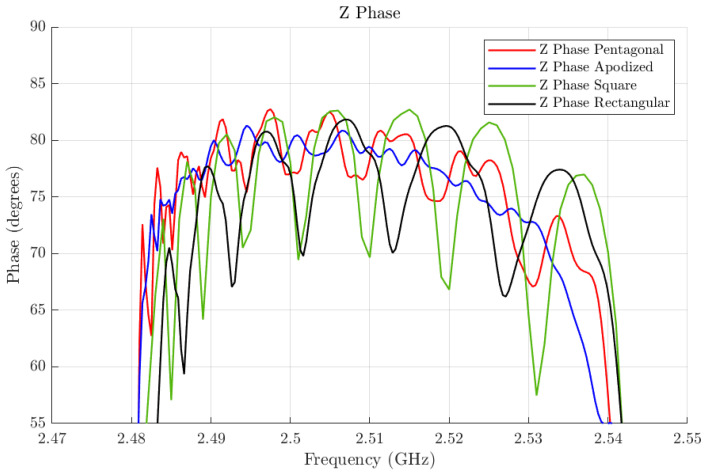
Comparison of the phase of the lateral impedance between the pentagonal SMR (in red), the apodized SMR (in blue), the square SMR (in green), and the rectangular SMR (in black).

**Figure 14 micromachines-16-00511-f014:**
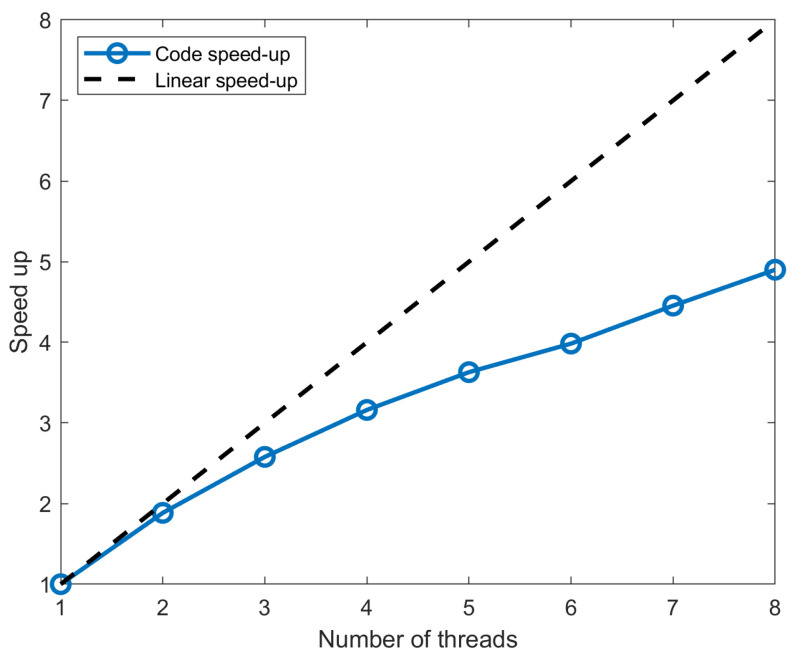
Plot of the speed-up as a function of the number of threads, compared to a linear speed-up.

**Table 1 micromachines-16-00511-t001:** Comparison of computational time and DoF between the FEM simulations and the proposed model with different meshing techniques.

Model	Structure	Time [DoF]
FEM	FBAR	3 h 47 min [576972]
Quasi-3D	FBAR	*Adaptive Mesh:* 18 s [6125]
		*Geometric with Dmax Mesh:* 32.5 s [8000]
		*Geometric with Dmin Mesh:* 32.5 s [8000]
		*Uniform Mesh:* 47 s [10125]
	SMR	*Square SMR:* 20.7 s [6400]
		*Rectangular SMR:* 28.2 s [7500]
		*Pentagonal SMR:* 18 s [6125]

## Data Availability

The data are not publicly available due to confidentiality.

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
