# Peer review of "Towards the Optimization of Apodized Resonators"

_micromachines, 2025, doi:10.3390/mi16050511_

Round 1

Reviewer 1 Report

Comments and Suggestions for Authors

In this manuscript, a development of Quasi-3D model for efficiently simulation BAW devices is presented. The idea is interesting, and the results are systematic with reasonable discussing. The authors can consider the following modifications:

  1. For different simulation examples, it would be better to present the schematic diagrams of geometric model for a better readability.
  2. For the simulation time in Table 1, does the Quasi-3D model simulation time contains the time consumed for obtaining the dispersion Curve? If not, it should also be added.
  3. For the standing patterns, it would be better to compare the results from COMSOL and Quasi-3D model.

Author Response

Comments 1: For different simulation examples, it would be better to present the schematic diagrams of geometric model for a better readability.

Thank you for your comment, we agree on your suggestion to improve readability. Therefore, we have added two figures (Figure 3 and Figure 9) where we show the diagrams for the FBAR and for the SMR geometries. The figures are located in pages 6 and 11.

Comments 2: For the simulation time in Table 1, does the Quasi-3D model simulation time contains the time consumed for obtaining the dispersion Curve? If not, it should also be added.

Response: We have accordingly modified the paragraph preceding Table 1 (page 14, lines 299-301) to emphasize this point.  We now clarify that the dispersion curve precomputation time is excluded, as it is a one -time cost per material stack. Moreover, once the parameters of the curve (clamb and r) are calculated, the computational time of the dispersion curve approximation becomes negligible. Nevertheless, we specify the computational time needed to compute the dispersion curve for a given stack.

              “The SMR computational cost is also introduced for three different structures, using only the adaptive mesh, since it is the one with the fastest convergence. The dispersion curve parameters (i.e., clamb, r) are precomputed from FEM or experimental data and reused across simulations. The execution time for 2D FEM simulations for the FBAR stack is of 10 seconds. This one-time cost is negligible compared to the runtime savings per type of geometry. Table 1 summarizes the results.”

Comments 3: For the standing patterns, it would be better to compare the results from COMSOL and Quasi-3D model.

Thank you for pointing this out. We agree that this comparison strengthens validation; therefore, we have modified Figure 7 (now Figure 8) to directly compare the standing wave patterns from COMSOL and our model, demonstrating strong agreement. In page 9 line 230, the reference to Figure 8a has been added. An additional comment has been made in page 10 lines 234-236 to discuss the additional subfigure.

              “Figure 8b shows the resonant modes obtained with COMSOL. As it can be seen, there is a strong agreement with the patterns obtained with the Quasi-3D models in symmetry and amplitude.

Reviewer 2 Report

Comments and Suggestions for Authors

This article is devoted to elaboration of the calculation methods for apodized acoustic resonators. At first, the Quasi-3D model is presented and some modifications is introduced in the model. Further, a wave propagation along the lateral dimensions of the resonator is discussed. Several corrections have been made to improve the accuracy of calculations and its adaptability to complex geometries of resonators. The model implementation is validated by comparing with FEM methods. The five-sided solid-mounted resonator is then described and optimized to minimize the lateral modes. An adaptive non-orthogonal mesh is proposed that better satisfy the shape of resonator. Two different meshing strategies is employed: adaptive meshing, using Chebyshev nodes, and geometric meshing, based on a geometric progression. The adaptive approach reduces computational cost, and higher resolution is achieved. The simulation time is essentially reduced as compared to FEM methods. The accuracy of the model is validated by the experimental measurements. An apodized design successfully diminishes the harmful effects of spurious resonances. The methods of calculations proposed can be useful for the other electro-acoustic devices.

The paper has essential novelty and presents new results. The topic of paper falls into the scope of Micromachines. The list of references is quite suitable. Only minimal corrections are necessary.

  1. The angular frequency ω appears in (1) but it explained only after (6). All designations have to be explained, but  f r and  f r001 appearing in (2) are not explained at all.
  2. Evidently, every designation through the article has to be related to one quantity. However, α in line 99 is a parameter corresponding to a type of the wave, and in line 144 α is the attenuation.
  3. The misprint has to be corrected in line 123.

Author Response

Comments 1: The angular frequency ω appears in (1) but it explained only after (6). All designations have to be explained, but fr and fr001 appearing in (2) are not explained at all.

We appreciate the reviewer’s attention to detail regarding symbol definitions. We have added a definition for the angular frequency immediately after (1), in page 3, line 98.

              “VL and VT are the phase velocities of the longitudinal and transverse waves, respectively, and ω is the angular frequency.

In response to your other comment, we have made the following improvements to ensure all variables and operations are unambiguous. The original equation (2) remains mathematically unchanged. However, we now explicitly state that both f and f001 are raised to the power of r before the subtraction and root operations. Moreover, f has been defined upon first use (page 4, lines 110-113).

              “where r ≈ 2, clamb is the estimated wave velocity, f is the operating frequency and f001 the cut-off frequency of the TE1 mode, also known as the piston mode. Here, (f r -f r001)1/r represents the dispersion relation’s frequency dependence, where f and f001 are first raised to the power of r before the subtraction and root operations.

Comments 2: Evidently, every designation through the article has to be related to one quantity. However, α in line 99 is a parameter corresponding to a type of the wave, and in line 144 α is the attenuation.

Thank you for pointing this out. We have clarified that θ (not α) corresponds to the wave type (page 3, line 99 and equation (1)), while α denotes the attenuation.

Comments 3: The misprint has to be corrected in line 123.

We apologize for the oversight. The misprint in line 123 has been corrected.